# Mechanisms and Pathophysiological Roles of the ATG8 Conjugation Machinery

**DOI:** 10.3390/cells8090973

**Published:** 2019-08-25

**Authors:** Alf Håkon Lystad, Anne Simonsen

**Affiliations:** Department of Molecular Medicine, Institute of Basic Medical Sciences and Centre for Cancer Cell Reprogramming, Institute of Clinical Medicine, Faculty of Medicine, University of Oslo, 1112 Blindern, 0317 Oslo, Norway

**Keywords:** ATG8, LC3, GABARAP, ATG5, ATG7, ATG16L1, autophagy, LAP

## Abstract

Since their initial discovery around two decades ago, the yeast autophagy-related (Atg)8 protein and its mammalian homologues of the light chain 3 (LC3) and γ-aminobutyric acid receptor associated proteins (GABARAP) families have been key for the tremendous expansion of our knowledge about autophagy, a process in which cytoplasmic material become targeted for lysosomal degradation. These proteins are ubiquitin-like proteins that become directly conjugated to a lipid in the autophagy membrane upon induction of autophagy, thus providing a marker of the pathway, allowing studies of autophagosome biogenesis and maturation. Moreover, the ATG8 proteins function to recruit components of the core autophagy machinery as well as cargo for selective degradation. Importantly, comprehensive structural and biochemical in vitro studies of the machinery required for ATG8 protein lipidation, as well as their genetic manipulation in various model organisms, have provided novel insight into the molecular mechanisms and pathophysiological roles of the mATG8 proteins. Recently, it has become evident that the ATG8 proteins and their conjugation machinery are also involved in intracellular pathways and processes not related to autophagy. This review focuses on the molecular functions of ATG8 proteins and their conjugation machinery in autophagy and other pathways, as well as their links to disease.

## 1. Introduction

Autophagy is crucial for the normal development of cells and organs, and its malfunction is linked to several human diseases. Macroautophagy (hereafter referred to as autophagy) is tightly regulated by several evolutionary conserved autophagy-related (ATG) proteins and involves sequestration of cytoplasmic material into a double-membrane autophagosome that fuses with the lysosome to allow cargo degradation (Figure 1). Autophagy is crucial for cell survival upon metabolic stress and for cell homeostasis by removal of dangerous cellular components, including dysfunctional organelles, intracellular microbes, and pathogenic proteins [1,2].

The molecular machinery involved in regulation of autophagy was initially discovered in yeast screens for survival upon nitrogen starvation and is largely conserved from yeast to man [3]. Several core ATG proteins form multi-subunit complexes, including i) the ATG1/Unc-51 like Kinase (ULK) complex, ii) the Vps34/class III phosphatidylinositol 3-phosphate kinase complex (PIK3C3) generating phosphatidylinositol 3-phosphate (PtdIns(3)P) at early autophagic structures, iii) the transmembrane protein ATG9 that shuttles to and from the sites of autophagosome formation and its trafficking machinery (ATG2 and ATG18/WD-repeat protein interacting with phosphoinositides 4 (WIPI4)) required for lipid transfer, and iv) the ubiquitin-like mammalian ATG8 (mATG8) homologues of the microtubule-associated protein (MAP1) light chain 3 (LC3) and γ-aminobutyric acid receptor associated proteins (GABARAP) subfamilies and their conjugation machinery. As this review will primarily focus on the mATG8 family members and their conjugation machinery, the other core ATG proteins will only be mentioned when linked to mATG8 function. The cellular and molecular mechanisms of the core ATG machinery have recently been thoroughly reviewed elsewhere [4,5].

Most ATG proteins are involved in the early stage of autophagosome formation and are themselves not degraded by autophagy. In contrast, the ATG8 proteins become directly conjugated to the lipid phosphatidylethanolamine (PE) in the autophagic membranes and remain bound throughout the pathway [6,7]. ATG8 proteins have therefore been the most widely used markers to study autophagosome biogenesis and trafficking [8]. Several studies have also addressed the cellular and pathophysiological importance of mATG8 conjugation to autophagy membranes. Pioneer studies in mice lacking ATG8 conjugation machinery components show that mATG8 lipidation, and autophagy, is important for neonatal survival and that tissue-specific knock-out (KO) of components of the conjugation machinery in adult mice can cause disease, including cancer and neurodegeneration, the latter caused by accumulation of protein aggregates in the mouse brain [9,10]. The main function of LC3 and GABARAP proteins in autophagy seems to be recruitment and scaffolding of proteins containing a so-called LC3-interacting region (LIR) (referred to as Atg8 Interaction Motif (AIM) in yeast). Some LIR-containing proteins facilitate autophagosome biogenesis (e.g., members of the ULK1 complex), while other LIR-containing proteins function as cargo-receptors to mediate specific targeting of cargo during selective autophagy [11]. Recently, it has however become evident that LC3/GABARAP proteins and members of their conjugation machinery also play a role in other membrane trafficking and signaling pathways. In this review we will discuss the role of mATG8 family members and their conjugation machinery in autophagy and other cellular pathways, as well as their roles in health and disease.

## 2. ATG8 Proteins and Their Conjugation Machinery

While there is only one Atg8 protein in yeast, the human ATG8 proteins are encoded by seven genes grouped into the LC3 (LC3A, LC3B, LC3B2, and LC3C) and GABARAP (GABARAP, GABARAPL1, GABARAPL2 (also called GATE-16)) subfamilies. While some animal lineages have lost members of the ATG8 family, many plants have gained genes encoding ATG8 proteins and can express more than ten different ATG8 proteins [12]. ATG8 proteins are members of the family of small ubiquitin-like modifier (UBL) proteins and contain in addition to the common ubiquitin-like fold two extra N-terminal α-helices [13,14]. The ATG8 proteins become covalently conjugated to the lipid phosphatidylethanolamine (PE) in a process that involves members of the ATG4 cysteine protease family and two ubiquitin-like conjugations systems that together regulate the three stages of LC3/GABARAP membrane conjugation; priming, PE-conjugation and de-lipidation (described in detail below and in Figure 2A,B).

### 2.1. Priming by ATG4 Cysteine Proteases

Newly synthesized pro-LC3 and pro-GABARAP proteins are recognized by ATG4 proteases that cleave their C-terminus to allow exposure of a free C-terminal glycine residue, which then becomes covalently attached to PE [15,16]. The processed forms of LC3/GABARAP proteins are referred to as LC3-I/GABARAP-I, while the PE-conjugated proteins are termed LC3-II/GABARAP-II (Figure 2A). There are four human ATG4 proteins (ATG4A, ATG4B, ATG4C, ATG4D) all shown to be able to cleave LC3/GABARAP proteins, but with different specificity towards the various LC3 and GABARAP family members as well as their presence as soluble or conjugated proteins [15,17,18,19,20] (Table 1). ATG4A and ATG4B have the closest pair-wise sequence identity and seem to facilitate both priming and de-lipidation of LC3 and GABARAP proteins. In contrast, ATG4C and ATG4D, which are more similar to each other than to ATG4A/B, appear to be specific for the de-lipidation of LC3/GABARAP proteins (described below) [18,19]. Rescue of mutant yeast strains lacking Atg4 by human *ATG4* indicate that ATG4-mediated processing of ATG8 is evolutionarily conserved [18]. It is also interesting to note that unprocessed forms of mammalian and yeast ATG8 proteins normally do not accumulate even when ATG4 levels are diminished by protein knockdown or drug treatment [21,22], indicating that this step is tightly regulated. Further studies are however needed to unravel the mechanisms underlying such regulation of LC3 and GABARAP protein levels.

### 2.2. LC3/GABARAP Lipidation

The PE-conjugation of LC3 and GABARAP proteins require ATG7 (E1) and ATG3 (E2), as well as the ATG12–ATG5–ATG16L1 complex (E3). Formation of the latter involves conjugation of ATG12 to ATG5, in a process mediated by ATG7 (E1) and ATG10 (E2), and further binding of the ATG12–ATG5 conjugate to an ATG16L1 dimer (Figure 2A,B-3,B-4). The various conjugation machinery proteins are described below.

#### 2.2.1. ATG7

ATG7 is the E1 like activating enzyme enabling both lipidation of ATG8 proteins and conjugation of ATG12 to ATG5. It is the only enzyme common to both conjugation pathways. The C-terminal domain of ATG7 is responsible for adenylation of the substrate (LC3/GABARAP or ATG12) by activating the C-terminal glycine of its substrate using ATP. This is followed by covalent attachment of the activated glycine residue to a catalytic cysteine residue in the ATG7 C-terminal through a thioester bond [32] (Figure 2A, Table 1). Finally, ATG7 promotes transfer of ATG8/ATG12 to the catalytic cysteine of the E2 conjugating enzymes ATG3 and ATG10, respectively. For yeast Atg7 it is shown that the protein forms a gliding bird-shaped dimeric architecture, through the dimerization of its C-terminal domain and further interaction of its N-terminal domain with either Atg3 or Atg10 [52,53,54,55,56]. Interestingly, Atg3 binds to one Atg7 and receives Atg8 from the catalytic cysteine of the opposite Atg7 molecule in the homodimer [53,54,55,56]. In the case of LC3 and GABARAP proteins the catalytic cysteine residue of the E2 enzyme ATG3 replaces that of ATG7, resulting in the transfer of LC3/GABARAP onto ATG3 via a trans-thioesterification reaction [24,27], while the E2 enzyme ATG10 has a similar role for ATG12 [57] (Figure 2A).

#### 2.2.2. ATG3

The main function of ATG3 is to assist in membrane conjugation of LC3 and GABARAP proteins. It does so through its E2-like function and by its direct binding to membranes, thus facilitating close proximity of LC3/GABARAP proteins to membranes.

The E2 function of ATG3 is mediated by a canonical ubiquitin conjugation domain (ATG3^core^) containing a ~100-residue long flexible region (ATG3^FR^) inserted between β2 and β3 of the ATG3 core [25,58]. The ATG3^FR^ contains a region responsible for interaction with ATG7 (region interacting with ATG7 (RIA7), aa ~157–181) [23], as well as a region interacting with ATG12 (RIA12, aa ~140–170) [25] (Figure 2A,B-1,B-2, Table 1). As these regions of interaction overlap at aa 157–170 it is not surprising that the E1–E2 (ATG7/ATG3) and E2–E3 (ATG3/ATG12–ATG5–ATG16L1) interactions are mutually exclusive [23,59]. Mutation of RIA7, required for the E1–E2 interaction, was found to severely impact GABARAP-loading of ATG3 [23], while RIA12 is crucial for recruitment of ATG12 and LC3B-II production [25]. Interestingly, the interaction between ATG12 and ATG3–RIA12 is structurally similar to the interaction of ATG8 family proteins with proteins containing a LIR motif [25]. As neither study included both LC3 and GABARAP proteins, it is likely that RIA7 and RIA12 are equally important for lipidation of both LC3 and GABARAP proteins. Importantly, the overlapping region favors interaction with the E3 complex [23], implying that the flow of the reaction cascade is biased towards the PE conjugation.

Membrane binding of ATG3 is facilitated by an N-terminal amphipathic helix (Figure 2A,B-6, Table 1), which was found to be important for its activity in LC3/GABARAP lipidation [28]. An N-terminal amphipathic helix in yeast Atg3 has also been proposed to be important for binding of Atg8 to PE containing membranes [60]. Additionally, positive interfacial lysine residues (located between the hydrophobic and polar face of the ATG3 amphipathic helix) provides affinity for negatively charged membranes [61]. Interestingly, in yeast it was found that acetylation of the N-terminal region of Atg3 promotes its Atg8 interaction and lipidation of Atg8 [62]. This was followed up in a later study where it was demonstrated that acetylation of Atg3 K19/K48 increases its interaction to PE-containing liposomes [63]. Whether a similar modification or other post-translation modifications also regulates the membrane binding affinity of human ATG3 remains to be seen. However, while the N-terminal amphipathic helix of ATG3 is required for LC3/GABARAP lipidation, in vitro and in vivo LC3/GABARAP lipidation experiments show that it is not sufficient for LC3/GABARAP lipidation in the absence of the ATG12–ATG5–ATG16L1 E3 complex [39].

Interestingly, the amphipathic helix of ATG3 is proposed to serve a curvature sensing function, as its membrane binding requires extensive lipid-packaging defects [28]. Thus, it is tempting to speculate that ATG3 facilitates ATG8 lipidation specifically at highly curved membranes, as seen at the limiting edge of the growing phagophore. As ATG8 proteins must be removed from the outer membrane before the final maturation of the organelle [28,64], continued lipidation on the autophagosome itself would be non-productive. The discovery that mATG8 proteins also can be conjugated to single membrane compartments with low curvature, does however indicate that ATG8 lipidation is tightly regulated through various mechanisms.

ATG3 is cleaved by caspase-8 in response to treatment with the death ligands Tumor necrosis factor-alpha (TNF-α) or Tumor necrosis factor-related apoptosis-inducing ligand (TRAIL), generating degradation products that are rapidly degraded by the proteasome [29]. Interestingly, the caspase-8 cleavage site (LETD, aa 166–169) overlaps with the ATG7 and ATG12 interacting region, suggesting that caspase-8-mediated processing of ATG3 is an efficient way of turning off autophagy under such pro-apoptotic conditions. Moreover, phosphorylation of ATG3 by protein tyrosine kinase 2 (PTK2) on Tyr203 upon DNA-damage inducing treatments was found to promote rapid degradation of ATG3 via an ubiquitin-proteasome-dependent pathway [65]. Prevention of ATG3 degradation during the DNA damaging process, lead to a significant decrease in cell proliferation in a mitotic catastrophe-dependent manner [65]. Thus, downregulation of ATG3 levels by posttranslational modifications and caspase cleavage under various conditions leading to cell death indicates that ATG3 is an important point for direct regulation of LC3/GABARAP lipidation and autophagy.

#### 2.2.3. The ATG12–ATG5–ATG16L1 E3-Like Complex

The main roles of the ATG12–ATG5–ATG16L1 E3-like complex in ATG8 lipidation is assumed to be membrane recruitment and activation of ATG3 to allow transfer of ATG8 proteins to their PE substrate [25,33,39,66,67,68,69]. ATG16L1 seem to specify the site of ATG8 lipidation [68], which is likely mediated by its interaction with specific membrane-bound proteins, including the PtdIns(3)P effector WIPI2b [35], the ULK1 complex component FAK family kinase-interacting protein of 200 kDa (FIP200) [36] and ubiquitin [37].

##### ATG12 Conjugation

ATG12 is an ubiquitin-like protein that is closest to the ATG8 family within the ubiquitin like protein kingdom [70]. To produce a functional E3-like complex the C-terminal glycine of ATG12 is conjugated to K130 of ATG5 in a process that requires the E1 ATG7 and the E2 enzyme ATG10 [32,71] (Figure 2A,B-3, Table 1). The ATG12–ATG5 conjugate then interacts with a dimer of ATG16L1 to facilitate membrane conjugation of LC3 and GABARAP proteins [66] (Figure 2A,B-4). Interestingly, some parasites and yeast species lack both ATG10 and the C-terminal glycine of ATG12 and instead form a non-covalent ATG12–ATG5 complex, which retains the ability to facilitate ATG8 lipidation [72].

It has also been reported that ATG12 can be directly conjugated to K243 of ATG3 in a process that requires the E1 ATG7 and the autocatalytic E2 activity of ATG3 (Figure 2A,B-10, Table 1) [26]. The ATG12–ATG3 conjugate is not required for starvation-induced autophagy, but rather display effects on mitochondrial mass and mitochondrial cell death pathways. Furthermore, the ATG3–ATG12 conjugate interacts with ALG-2-interacting protein X (ALIX, also known as PDCD6IP), affecting multivesicular body (MVB) distribution, exosome biogenesis and viral budding [73] (Figure 1). There is no de-conjugation enzyme for ATG12, thus conjugates are stable.

##### The ATG12–ATG5 Conjugate

Structural and mutational analyses suggest that both ATG12 and ATG5 are directly involved in the E3 activity of the complex through residues that are assembled into a continuous surface patch upon formation of the ATG12–ATG5 conjugate [31]. This patch contains the conjugation site and C-terminal residues of ATG12, including A138, W139, and G140. Other residues such as V62, G63, L92, Q106, and S107, located on various segments of ATG12, and the ATG5 residues K138, N143, M145, Q146, H150, and I168 also participate in this patch [31]. Moreover, a non-canonical AIM in yeast Atg12 is required for binding to Atg8 and scaffold formation by the Atg12–Atg5–Atg16 complex on phagophores [33]. This motif may correspond to W139 and V62 in human ATG12, but any functional role of this in human proteins remains to be studied (Figure 2A,B-5, Table 1).

In addition to the interaction with ATG16L1 and the essential E3-like function in membrane conjugation of LC3/GABARAP proteins, the ATG12–ATG5 conjugate has been found to interact with the lysosomal located tectonic β-propeller repeat containing 1 (TECPR1), which regulate autophagosome–lysosome fusion and is important for targeting bacterial pathogens for autophagy [74,75]. Binding of ATG5 to TECPR1 is mutually exclusive to its binding to ATG16L1, as both TECPR1 and ATG16L1 bind the same site in ATG5 [34,74]. Interestingly, the ATG12–ATG5 interacting region (AIR) of TECPR1 normally conceals a Pleckstrin homology (PH) domain in TECPR1 that is exposed upon its interaction with ATG5, leading to binding of the PH domain to PtdIns(3)P [74]. Thus, ATG5 seems to regulate several steps of the autophagy pathway, however the mechanisms underlying the selectivity of ATG12–ATG5 binding to either ATG16L1 or TECPR1 in time and space are not know.

##### ATG16L1

While ATG16L1 is dispensable for the conjugation of ATG12 to ATG5, its binding to the conjugate and dimerization is essential for lipidation of ATG8 proteins [76,77].

Two crystal structures of the ATG12–ATG5 conjugate bound to a small N-terminal part of ATG16L1 (aa 1–43 or aa 1–69) exist [31,34]. These show that ATG16L1_11–43_ interacts with a surface of ATG5, consisting of two ubiquitin-like fold domains (referred to as UFD-1 and UFD-2), in an α-helical conformation with a slight kink in the middle [31] (Figure 2A,B-4). ATG12 binds to the opposite side of ATG5 compared to ATG16L1_11–43_, and no contacts between ATG12 and ATG16L1_11–43_ are seen [31]. The interaction between ATG5 and ATG16L1 is mainly mediated by the ATG5 (Five)-Interacting Motif (AFIM) in ATG16L1 stretching from aa 13–28, although the remaining part of the helix also seems to contribute to its interaction with ATG5 (Figure 2B-4, Table 1). The C-terminal portion of ATG16L1_1–69_ contains several hydrophobic residues situated toward the interface with ATG5, such as F32, I36, T39, and L43 and it was found that these hydrophobic amino acids allow formation of a crystallographic homodimer [34]. The formation of a dimer in solution was further verified by size exclusion chromatography plus multi-angle light scattering (SEC-MALS) analysis of ATG5–ATG16L1_1–69_. Thus, the N-terminal helix of human ATG16L1 is both sufficient for interaction with ATG5 and for self-dimerization, even without the middle coiled-coil region.

Interestingly, we recently found that when the hydrophobic amino acids of the second part of the ATG16L1 N-terminal helix region (referred to as helix 2, aa 29–46) were exposed to liposomes, they provide membrane binding as an amphipathic helix [39] (Figure 2B-7, Table 1). This membrane interaction is not required for recruitment of the E3 ligase complex to membranes in vivo, but is essential for efficient LC3/GABARAP lipidation both in vitro and in vivo. Thus, it is likely that membrane binding by ATG16L1 helix 2 positions and stabilizes the other members of the conjugation machinery in close proximity to the membrane surface to allow lipidation to occur. This could be facilitated by the combined membrane binding affinity of the ATG3 and ATG16L1 amphipathic helices or by the assembly of oligomeric ATG12–ATG5–ATG16L1 structures, which upon membrane insertion of helix 2 could cause membrane deformations so that ATG3 can insert its amphipathic helix.

Membrane recruitment of ATG16L1 is also facilitated by its binding to WIPI2b (where mutation of ATG16L1 E226 and E230 abolish the interaction) [35] and FIP200 (aa 239–246) [36,37,38] (Figure 2B-8, Table 1). The coiled-coil domain of ATG16L1 (aa 78–230) provides dimerization of the full length protein [76], and harbor a binding site for Ras-related protein Rab-33B (RAB33b, aa 200–265) [78] as well as residues (I171, K179, and R193) that seem to be important for recruitment of PtdIns(3)P containing liposomes to ATG16L1-coated beads (Figure 2B-8, Table 1) [40]. The linker region connecting the coiled-coil domain to the C-terminal WD40 repeats defines the difference between the two isoforms of ATG16L1 found in humans, the α and β isoform. The α isoform lacks an exon compared to the β isoform and are thus missing aa 266–284 of the β isoform. Also, a γ isoform has been found in mice, having a 16 amino acid insert after aa 284 of ATG16L1 β. Interestingly, the β isoform specific region (aa 266–284) is highly phosphorylated (phosphosite), where S278 has been verified as a substrate for the ULK1 kinase in response to infection or starvation [79] (Figure 2B). Moreover, this region was shown to bind membrane in vitro and to be important for localization of ATG16L1β to swollen endosomes in cells [39].

The C-terminal WD domain of ATG16L1 is not found in yeast Atg16 and is thus dispensable for conventional autophagy. However, this part of ATG16L1 is required for lipidation of LC3 to single- membrane compartments in other processes than autophagy, including LC3-associated phagocytosis (LAP) and LC3 lipidation to perturbed endosomes in response to various drugs possessing lysosomotrophic or ionophore properties [39,41] (described in more detail below). During LAP, this domain seems to facilitate recruitment of ATG16L1 through a direct interaction with ubiquitin [37]. Interestingly, the residues F467 and K490, contained within a pocket on the top of the WD domain, are important for such non-canonical roles of ATG16L1 [41] (Table 1), but any specific interaction partners are yet to be discovered.

It is interesting to note that the ATG12–ATG5 conjugate also forms a complex with the ATG16L2 isoform, which shares the same domain structure as ATG16L1, but does not bind to WIPI2b and is unable to support canonical autophagy [35,80]. As cells lacking ATG16L1 are unable to execute LAP and LC3 lipidation at swollen endosomes [39,41] it is unlikely that ATG16L2 can substitute for ATG16L1 in lipidation of LC3 to single membrane structures. It will be exciting to learn the function of the ATG12–ATG5–ATG16L2 complex.

### 2.3. LC3/GABARAP De-Lipidation

LC3/GABARAP proteins conjugated to the outer autophagosome membrane are removed by ATG4-mediated cleavage before fusion with the endo-lysosomal pathway [64,81]. LC3 positive autophagic membranes formed during starvation are known to persist for approximately 10–20 min after LC3 starts to associate with these structures [6,82,83]. During this time, LC3/GABARAP bound to the outer membrane are important for transport and maturation of autophagosomes by binding to different LIR-containing proteins, e.g., FYVE and coiled-coil domain-containing protein 1 (FYCO1) that facilitates interaction of autophagosomes to microtubule plus-end directed motors [84] and the PH domain containing protein family member 1 (PLEKHM1), which mediates fusion of autophagosomes to lysosomes [85]. LC3/GABARAP de-lipidation must therefore be tightly regulated in time and/or space to prevent their premature removal.

Exactly how ATG8 proteins are allowed to stay on the membrane until their job is done is not known. It has been proposed that an elevated level of reactive oxygen species (ROS) is needed to facilitate the productive assembly of autophagosomes by suppressing ATG8 de-lipidation at the autophagosome assembly site. Indeed, it was shown that antioxidants prevent localization of ATG8 proteins to autophagic membranes, which may indicate that ROS, especially H_2_O_2_ that accumulate in cells during starvation, are required for autophagy [86]. In line with this model, ATG4A and ATG4B contain oxidation-prone cysteine residues in their active sites, which render these proteins sensitive to ROS [86]. Oxidative stress can also cause direct oxidation of ATG3 and ATG7, which was found to inhibit their activity in LC3 lipidation [87].

However, basal autophagy appears to occur more or less constitutively in mammalian cells, and autophagosome formation is not limited to periods of prolonged stress. Thus, insight into how membrane conjugated LC3 and GABARAP proteins persist long enough to support autophagosome maturation when ATG4 activity is otherwise normal has been an area of intense investigation. Recent studies have identified conserved C-terminal LIR motifs in all ATG4 family members. In addition, ATG4B has a conserved N-terminal LIR that engages a non-substrate LC3 protein in the co-crystal of the protease and substrate [47] (Table 1). The C-terminal LIR of ATG4B was found to have two roles; 1) to stabilize the pool of soluble GABARAP-family proteins by preventing their degradation by an autophagy-independent turnover pathway and 2) to support de-lipidation, as demonstrated for de-lipidation of GABARAPL1–PE (reduced by a factor of ~4) and LC3B-PE (completely abolished) [19,42]. Mutation of the ATG4B C-terminal LIR did however not affect priming of LC3/GABARAP proteins, The N-terminal LIR of ATG4B had no effect on priming and only a modest effect on de-lipidation with a 2-fold reduction on LC3B–PE and insignificant effect on GABARAPL1–PE [19].

Phosphorylation of ATG4B at S383 and S392 was found to increase its hydrolase activity toward membrane-conjugated LC3 and be important for proper autophagic flux, while LC3B priming seemed not affected by ATG4B phosphorylation in cells [88]. Interestingly, ULK1 was found to phosphorylate ATG4B on S316, thereby inhibiting its catalytic activity, while the phosphatase PP2A-PP2R3B could remove this inhibitory phosphorylation [89]. Thus, a tightly regulated phospho-switch seems to control the activity of ATG4B and likely of other ATG4 proteins.

ATG4C and ATG4D are unable to mediate priming of any tested ATG8 proteins, but ATG4C display a modest ability to de-lipidate GABARAPL2 [19]. The general lack of activity of ATG4C and ATG4D towards LC3 and GABARAP proteins is consistent with work suggesting that these proteases are ordinarily maintained in an auto-inhibited state by the presence of an amino-terminal inhibitory domain [19,50]. This domain can be removed by caspase cleavage at a conserved DEVD motif present in both proteins [50]. Indeed, “activated” ATG4C and ATG4D proteins gain activity, but only towards the lipidated form of LC3/GABARAP proteins. Thus, ATG4C and ATG4D are de-lipidation-specific enzymes [19].

Caspase cleavage of ATG4C and ATG4D also expose a mitochondrial targeting sequence (MTS) at their N-terminal region, as shown by the ability of the N-terminal 30–40 aa to target green fluorescent protein (GFP) to mitochondria [50,51] (Table 1). The mitochondria localized ATG4D was protected from proteinase K, indicating it is imported into the mitochondria [51]. Additional cleavage of the ATG4D MLS by mitochondria metalloproteinases caused a further truncated product, but it should however be noted that expression of a caspase-resistant ATG4D mutant (DEVD mutated to DEVA) gave a protein of similar size, in addition to the normal full length version, indicating that caspase cleavage of ATG4D may not be necessary for import into mitochondria and subsequent processing [51].

ATG4 activity can also be regulated by modulation of ATG4 protein levels. Both ATG4A and ATG4B was found to undergo calpain 1 cleavage [90] and the level of ATG4B to be regulated by Ring finger protein 5 (RNF5)-mediated ubiquitination and degradation [91]. In line with a decreased activity of ATG4B causing increased autophagy, RNF5 deficient mice were more resistant to group A *Streptococcus* infection, a known cargo for selective autophagy [91].

## 3. Role of LC3/GABARAP Proteins in Autophagy

Atg8 and LC3 were the first specific autophagosome markers described and have been widely used in the field of autophagy to analyze autophagosome formation and autophagic flux (autophagic degradation activity), using various microscopy-based methods (e.g., immunofluorescence staining for endogenous LC3 or quantification of GFP–LC3 structures) or by Western blot analysis of lipidated LC3 (as LC3-II migrates faster than LC3-I). In the absence of other specific autophagy markers, it has been difficult to investigate the pathway and many studies have concluded about autophagy levels based on the levels of membrane conjugated LC3 observed. Such conclusions are likely valid in most cases, but several new discoveries in the field indicate that LC3 should not be used as a general indicator of autophagy. Most importantly, recent studies have found that autophagosomes can still form in cells depleted of components of the LC3 conjugation machinery (such as ATG3, ATG5 or ATG7) [92] or of all LC3 isoforms (LC3A/B/C) [93,94]. Moreover, LC3 positive structures may not be autophagosomes. As LC3 interacts with ubiquitin-binding autophagy receptors (e.g., p62), it will be recruited to protein aggregates accumulating in cells with compromised autophagy [8,95]. Another potential pitfall by using LC3 as a marker for autophagy, is the recently recognized conjugation of LC3 to membranes not involved in autophagy, including phagosomes and perturbed endosomes (described in more detail below). Thus, although LC3 is still our best marker for autophagy, it is important to include proper controls when using it.

### 3.1. Autophagosome Biogenesis

The soluble N-ethylmaleimide-sensitive factor attachment protein receptor (SNARE) protein syntaxin 17 (STX17) was found to be required for fusion of autophagosomes with the endosome/lysosome [96]. Importantly, STX17 localizes to the outer membrane of completed autophagosomes through a unique C-terminal hairpin structure, providing an explanation as to why the lysosome does not fuse with early autophagy structures, but also an alternative marker to study autophagosome formation. Interestingly, only two minutes after recruitment of STX17, small lysosomes start to fuse with the autophagosomes, which become lysotracker positive, suggesting acidification of the space between the inner and outer autophagosomal membrane. Approximately seven minutes later the autophagosomal matrix was entirely acidified, suggesting that the inner autophagosomal membrane is degraded [92]. When analyzing STX17-positive autophagosome biogenesis in ATG conjugation-deficient mouse embryonic fibroblasts (MEFs), it was found that autophagosome-like structures that were able to fuse with lysosomes formed, although at a reduced rate, but that degradation of the inner autophagosomal membrane was significantly delayed and by extension also cargo degradation [92]. Thus, the conjugation machinery, and by extension LC3 and GABARAP proteins, are likely important for efficient closure of the phagophore by fission of the inner and outer autophagosomal membrane of the phagophore edge.

Another study investigated the functional significance of deleting LC3 and GABARAP subfamily proteins, either individually (LC3 or GABARAP triple KO (TKO)) or together (LC3/GABARAP hexa KO) in HeLa cells. In line with the previously described paper, they found that autophagosomes form in the hexa KO cells, but that they are smaller and unable to fuse with lysosomes [93]. Thus, LC3 and GABARAP proteins are likely important for regulation of autophagosome size, which has also been suggested in previous studies [82,97,98,99]. It is not clear whether the apparent different ability of autophagosomes to fuse with lysosomes found in these two studies are due to use of different experimental systems or if deletion of all ATG8 homologs have other effects than deletion of critical conjugation machinery components.

The mechanisms underlying the function of LC3/GABARAP lipidation in autophagosome closure is not completely understood, but seem to be specific to the GABARAP subfamily. While depletion of all LC3s had no effect on starvation-induced autophagic flux, this was inhibited by depletion of all GABARAPs [93,94]. The mechanistic explanation for this is not clear, but it is interesting to note that several LIR-containing proteins are specific GABARAP interactors [100]. Among those are the ULK1 kinase and ATG13, a member of the ULK1 complex, and it was found that the LIR motif is required for starvation-induced association of ULK1 with autophagosomes [101]. Interestingly, a recent paper identified GABARAP and GABARAPL1 as positive regulators of ULK1 activity and phagophore formation in response to starvation, while LC3B and LC3C was found to negatively regulate ULK1 activity and phagophore formation [102].

Moreover, it was recently demonstrated that an interaction of ATG2A/ATG2B with GABARAP is important for phagophore closure [103], in line with the observed accumulation of unclosed autophagic structures containing several autophagy proteins in cells lacking ATG2 [104,105,106]. ATG2 is a lipid transfer protein that seem to tether the tip of the expanding phagophore with the endoplasmic reticulum (ER) to facilitate lipid transfer and autophagosome biogenesis [107,108,109]. Recruitment of Atg2/ATG2B and its binding partner Atg18/WIPI4 to the phagophore was found to require Atg9 [110] and Trafficking protein particle complex subunit 11 (TRAPPC11) and depletion of TRAPPC11 phenocopied that of ATG2 deletion [111]. The endosomal sorting complexes required for transport (ESCRT) III components are also necessary for autophagosome closure [112,113], but their link if any to GABARAP has not been revealed. LC3 and GABARAP proteins have also been reported to possess membrane tethering and fusogenic properties [98,114,115]. Whether this is a function required for fission of the phagophore membrane is still not clear, but it has been proposed that GABARAP proteins could bridge some portion of the fission pore to facilitate organelle closure. Consistent with such an idea it was shown that lipidated GABARAPL1 accumulate at sites of high curvature and membrane–membrane apposition, making the open fission pore of a forming autophagosome an ideal site of function [115].

Membrane recruitment of the specific GABARAP interactor PLEKHM1, which facilitates autophagosome-lysosome fusion [85,100] was abolished in cells lacking all GABARAPs [93], providing an explanation for the lack of autophagosome–lysosome fusion in these cells. Another possible role for GABARAP in autophagosome maturation is by recruiting the phosphatidylinositol 4-kinase 2-alpha (PI4K2A) to autophagosomes, leading to generation of PtdIns4P that promotes autophagosome fusion with the lysosome, likely by recruiting elements of the membrane docking and fusion machinery [116]. Using an electron microscopy technique that labels PtdIns4P on the freeze-fracture replica of intracellular biological membranes, it was found that PtdIns4P localizes on the cytoplasmic, but not the luminal leaflet of both the inner and outer autophagosome membrane [117]. Moreover, PtdIns4P colocalizes with the late endosomal RAB7, a binding partner of PLEKHM1, further suggesting an important role of GABARAP in coordination of autophagosome-lysosome fusion events.

### 3.2. Selective Autophagy

So, what is the function of the LC3 isoforms? Interestingly, while starvation-induced autophagy and E3 ubiquitin-protein ligase parkin (PRKN) mediated mitophagy seem to occur independently of LC3 subfamily proteins, basal autophagy was reduced in the HeLa LC3 TKO, as evident by an accumulation of the autophagy receptor p62/Sequestosome 1 (SQSTM1). It should however be noted that the effect on starvation-induced autophagy and mitophagy was greater in the hexa KO cells than in the GBRP TKOs, suggesting that LC3s can partially compensate for the lack of GABARAPs. Moreover, all GABARAP isoforms (GABARAP, GABARAP-L1, GABARAP-L2) were able to rescue autophagy in hexa KO cells [93], suggesting these proteins are redundant.

Basal autophagy is defined as the level of autophagy seen under normal, non-induced conditions and serves an important house-keeping function by removal of damaged or dysfunctional organelles and cellular proteins. Such selective turnover of specific cargo is generally referred to as selective autophagy, in contrast to the seemingly “random” or “bulk” uptake attributed to starvation-induced autophagy. Cargo for selective autophagy can range from specific proteins, such as fatty acid synthase [118] and ferritin [119], poly-carbohydrates such as glycogen [120] or multi-protein aggregates [121], to parts of organelles as the ER and up to entire organelles such as mitochondria [122] or invading pathogens [2].

Selective autophagy depends on autophagy receptors, which are cargo-binding proteins that bind to LC3/GABARAP proteins via a LIR and get themselves degraded within lysosomes together with the cargo [123]. Yeast Atg19 was the first autophagy receptor identified, being important for selective targeting of a precursor form of protein aminopeptidase I (prAPI) to the vacuole in a process referred to as the cytoplasm-to-vacuole (Cvt) pathway [124,125,126]. Although autophagy receptors were originally identified as cytosolic proteins, such as p62, Next to BRCA1 gene 1 protein (NBR1) or Nuclear dot protein 52 (NDP52) binding to ubiquitinated cargo, it was later found that cargo-specific membrane-bound proteins also can function as autophagy receptors, including proteins in mitochondria (BCL2/adenovirus E1B 19 kDa protein-interacting protein 3 (BNIP3), BCL2/adenovirus E1B 19 kDa protein-interacting protein 3-like (BNIP3L) and FUN14 domain-containing protein 1 (FUNDC1)) [122] and the ER (Family with sequence similarity 134 member B (FAM134b), Reticulon-3 (RTN3), cell-cycle progression gene 1 (CCPG1) and Testis-expressed protein 264 (TEX264)) [127]. Thus, autophagy receptors can either facilitate cargo degradation by mediating interaction between a cargo protein (often ubiquitinated) and LC3/GABARAP in the autophagosomal membrane or by direct recruitment of LC3/GABARAP to the cargo.

Consequently, an important function for membrane conjugated LC3 and GABARAP is to act as attachment points for various receptor proteins. A landmark study from 2010 identified the interactome of all LC3 and GABARAP proteins, which was found to comprise 67 proteins [128]. Many of these proteins have later been found to contain a LIR, which is typically composed of a core W/F/Y-x_1_-x_2_-L/V/I motif, where the respective aromatic and hydrophobic amino acids interact with two hydrophobic pockets on the LC3/GABARAP proteins [11,129]. Many proteins have acidic residues N-terminal to the core motif, which seem to increase the affinity of their interaction with LC3/GABARAP [11]. Interestingly, not all LIR motifs contain acidic residues at the start, and it has been found that many LIRs instead have a phosphorylatable residue upstream of the core LIR residues that upon phosphorylation introduces a negative charge akin to an acidic residue that can enhance binding to the basic (R10/R11/K51) patch in LC3 [130]. This was first revealed for the receptor protein optineurin (OPTN) that becomes phosphorylated by the kinase TANK-binding kinase 1 (TBK1) in response to cellular *Salmonella* infection and it was shown that preventing this phosphorylation reduced the efficiency of LC3 targeting to invading bacteria [131,132]. The concept of LIR phosphorylation has also been observed with the mitophagy receptors BNIP3L, FUNDC1 and BNIP3 [133,134,135,136]. BNIP3L contains two serine residues upstream of the LIR motif (NSSWVEL), which when phosphorylated enhances LC3B binding affinity up to 100-fold [133].

Since the initial identification of the LIR it has become clear that certain residues in the core LIR motif confer specificity towards various LC3/GABARAP subfamilies or subfamily members. Interestingly, most autophagy receptors possess an LIR motif which allows their direct binding to LC3. In contrast, most autophagy adaptor proteins, which are important for autophagy, but not degraded by autophagy, have GABARAP-specific LIRs. Thus, LC3 proteins are believed to be particularly important for recruitment of cargo upon selective autophagy. To be able to further analyze the localization and function of the different endogenous LC3/GABARAP proteins, specific fluorescence-based ATG8 sensors have been developed [137,138]. Such sensors are based on the different LIR- peptide specificities of the various LC3 and GABARAP proteins and will likely provide novel insight into the cellular localization of these proteins.

The simple linear model of selective autophagy, in which cargo-specific receptors facilitate recruitment of LC3-containing phagophores to facilitate sequestration of the cargo into autophagosomes, has recently been challenged by several elegant studies showing that receptor proteins, as p62 and NDP52, rather recruit the ULK1 complex to initiate de novo phagophore formation directly at the cargo, independently of LC3 [139,140,141]. p62 and NDP52 were found to interact directly with the ULK1 complex subunit FIP200 to facilitate cargo-recruitment of the ULK1 complex, which again led to recruitment of other core ATG proteins and de novo phagophore formation. This binding seems to be regulated by phosphorylation of the receptors, TBK1 in the case of NDP52 [140,141]. Interestingly, the interaction of p62 with FIP200 was outcompeted by LC3B [139], suggesting a sequential and directional order of p62 binding partners during phagophore formation in selective autophagy. In line with such a de novo model of phagophore formation, LC3/GABARAP proteins were found to be recruited to damaged mitochondria independent of their binding to autophagy receptors further facilitating ubiquitin-independent recruitment of OPTN and NDP52 to growing phagophore membranes via the LIR motif, which again would amplify PTEN-induced putative kinase protein 1 (PINK1)/PRKN-dependent mitophagy [142]. Thus, de novo building of the phagophore around a specific cargo seems to involve an intricate and highly regulated network of protein interactions, including a LC3/GABARAP-dependent positive feedback loop. It still remains elusive where the lipids/membranes for such cargo-specific phagophore formation come from and whether the mechanisms of phagophore formation differ from starvation-induced autophagy. The origin of the phagophore membrane and the mechanisms involved in autophagosome biogenesis have been extensively described in several other review articles (for recent reviews see e.g., [5,143]) and will not be further discussed here.

### 3.3. Regulation of LC3/GABARAP Proteins

Several post-translational modifications of LC3B have been identified. Acetylation of K49 and K51 seem to mediate LC3B localization to the nucleus, as deacetylation of these residues was found to facilitate nuclear export and the function of LC3B in autophagy [144]. Interestingly, phosphorylation of LC3B S12 appears to negatively regulate its function in autophagy, as inducers of autophagy caused dephosphorylation of endogenous LC3 and a non-phosphorylatable S12A mutant exhibited enhanced puncta formation [145]. Although not yet demonstrated, it is possible that phosphorylation of LC3B S12 prevents its interaction with LIR-containing proteins, as acidic residues upstream of the LIR motif bind the positively charged R10/R11 residues in LC3B. The mechanisms involved in regulation of LC3B S12 phosphorylation need to be further elucidated, but it was demonstrated that while protein kinase A (PKA) could phosphorylate LC3B S12 [145], the mitogen-activated protein (MAP) kinase MAPK15/ERK8 reduced S12 phosphorylation and stimulated LC3B lipidation [146]. Recently, it was shown that TBK1 also phosphorylates several LC3 and GABARAP subfamily proteins, except LC3B and GABARAP, and further investigation revealed that TBK1 indeed can regulate autophagy by phosphorylating LC3C on S93 and S96 and GABARAPL2 on S87 and S88 [46]. These phosphorylation events impede de-lipidation of LC3C and GABARAPL2 by ATG4A and ATG4B, thus protecting them from premature removal from forming autophagosomes [46]. Thus, post-translational modifications of LC3/GABARAP proteins seem to regulate their function in autophagy and provide directionality for autophagosome maturation.

## 4. Non-Conventional Roles of ATG8s and Their Conjugation Machinery

In this review we use the phrasing “non-conventional roles” to describe non-autophagic roles of ATG8 proteins and their conjugation machinery in the cell. Although the term non-canonical autophagy is often applied to such alternative functions of LC3/GABARAP proteins, non-canonical autophagy is also used to describe autophagic degradation in the absence of various core ATG proteins, including the autophagic conjugation machinery [147].

### 4.1. Lipidation of ATG8s to Single-Membrane Compartments

During canonical autophagy, ATG8 proteins are conjugated to PE in the double-membrane autophagosome. However, ATG8 proteins can also be conjugated to various single-membrane compartments in response to different stimuli. The function(s) of ATG8 proteins on single membranes are however generally unclear.

A common feature of non-conventional LC3/GABARAP lipidation on single-membranes seems to be that it can occur independently of nutrient status and upstream regulators of autophagy, including components of the ULK1 complex and the ATG14L-containing Vps34/PIK3C3 complex 1, although it generally requires the conjugation machinery [148,149]. Remarkably, while canonical autophagy can occur independently of the C-terminal WD domain of ATG16L1, this domain seems essential for LC3/GABARAP lipidation to single-layered membranes [39,41]. Examples of LC3/GABARAP lipidation to single membranes include translocation of LC3 to damaged Golgi membranes [150], relocation of LC3 to vesicles in the endo-lysosomal system following various endocytic engulfment events, such as LC3-associated endocytosis (LANDO) [151], entosis (live-cell cannibalism) [149,152], micropinocytosis [41,149], and phagocytosis (referred to as LC3-associated phagocytosis, LAP) [153,154,155,156] (Figure 1). It has been proposed that LC3 lipidation promotes maturation of phagosomes during LAP, but the underlying molecular mechanisms are not clear [148,149].

### 4.2. Induction of Non-Conventional ATG8 Protein Lipidation

ROS production was found to be a prerequisite for LC3 lipidation to the phagosomal membrane during LAP [148]. The nicotamide adenine dinucleotide phosphate (NADPH) oxidase-2 (NOX2) complex was stabilized by PtdIns(3)P, leading to generation of ROS for ATG8 recruitment to the phagosome [148]. The PtdIns(3)P in turn was produced by the PI3KC3 complex II, containing UV radiation resistance-associated gene protein (UVRAG) and Rubicon, but not ATG14L and Activating molecule in BECN1-regulated autophagy protein 1 (AMBRA1) [148]. The question then is how do ROS cause LC3 lipidation during LAP? Interestingly, ROS accumulation in phagosomes alters transmembrane ionic balances and promotes water influx [157], effects that have been shown to cause non-conventional LC3/GABARAP lipidation on other membranes as well [158].

Moreover, several reports on non-conventional LC3/GABARAP pathways focus on the close link to V-ATPase function [158,159,160]. It is proposed that the V-ATPase complex actually detects damage to membrane compartments by sensing changes in the proton gradient, triggering a conformational change, which results in recruitment of the conjugation machinery to promote LC3/GABARAP lipidation at the perturbed membrane. Indeed, it was recently shown that the WD domain of ATG16L1 facilitates an interaction with the V-ATPase upon bacteria-induced vacuolar membrane damage in *Salmonella* infected cells, leading to LC3 lipidation on the *Salmonella* containing vacuole (SCV) and a reduction in the bacterial burden [159].

### 4.3. Role of ATG8 Proteins and Their Conjugation Machinery in Secretion and ER Export

The autophagic conjugation machinery has also been implicated in several secretory events, including von Willebrand factor (vWF) secretion in endothelial cells (ATG5 and ATG7 dependent) [161], neuropeptide Y (NPY) secretion from neuroendocrine cells (ATG16L1 dependent) [162], mucin 5AC (MUC5AC) secretion from airway endothelial cells (ATG16L1, ATG5, and ATG14L dependent) [163] and lysozyme secretion in Paneth cells (ATG16L1 dependent) [164]. Upon *Salmonella typhimurium* infection, lysozyme secretion become resistant to Brefeldin A treatment, suggesting it by-passes ER to Golgi transport [164] (Figure 1). Importantly, these Golgi independent secretory lysozyme granules become surrounded by a LC3 positive double membrane. Moreover, secretion of lysozyme becomes sensitive to mutations in ATG16L1 and to treatment with the autophagy inhibitor 3-methyl adenine (3-MA), corroborating the importance of the autophagy machinery in this non-conventional secretory pathway. Interestingly, autophagy-dependent secretion of lysozyme does not involve lysosomal degradation, as treatment with chloroquine, which disrupts lysosomal acidification, has no effect on lysozyme secretion.

Secretion of leaderless cytoplasmic proteins is often referred to as secretory autophagy as it utilizes autophagic core components (Figure 1). Substrates include IL-1β [165,166,167], high-mobility group box 1 (HMGB1) protein [165,168] and insulin-degrading enzyme (IDE) [169]. IDE is one of the major proteases of amyloid beta peptide (Aβ), presumed to be a causative molecule in Alzheimer disease (AD) pathogenesis. Experiments with astrocytes, the main IDE secreting cells, showed that IDE secretion upon stimulation with amyloid beta peptide required the activity of ATG5, RAB8A, and Golgi reassembly stacking protein (GORASP). Additionally, autophagic inhibitors like 3-MA reduced secretion, while rapamycin, an inducer of autophagy, promoted secretion. Secretion of IL-1β, particularly that from non-immune cells, was found to occur through a similar pathway that required ATG5, ATG16L1, LC3B, GABARAP, Golgi reassembly-stacking proteins (GRASPs), and RAB8A with IL-1β found in LC3-positive structures [165,166,167,170]. The similar requirements found for secretion of these substrates argues for a common pathway that uses an ”autophagosome-like” intermediate. However, IL-1β can also be secreted through Gasdermin D (GSDMD) pores in the plasma membrane [171,172,173]. GSDMD is processed by caspase during inflammation, generating a truncated product that is efficiently recruited to the plasma membrane where it can form a 16-fold-symmetry pore that allows secretion of IL-1β [171,173]. Recently it was also shown that IL-1β accumulates at PtdIns(4,5)P_2_-enriched plasma membrane ruffles, further indicating that IL-1β can be secreted both through GSDMD-dependent and -independent pathways [174].

Another interesting non-conventional role for ATG8 proteins is the discovery that lipidated LC3C was required, in cooperation with TECPR2, to maintain functional ER exit sites (ERES) [175]. This requires a LIR-dependent interaction between TECPR2 and LC3C, thereby providing an anchor for TECPR2 on endomembranes, allowing it to function as a multifunctional scaffold protein for SEC24D in an early secretory pathway, acting as a positive regulator of coat protein complex II (COPII)-dependent ER export [175].

### 4.4. Role of ATG8 Proteins and Their Conjugation Machinery in Endosomal Microautophagy

Recently, a starvation-induced response referred to as endosomal microautophagy was found to cause the rapid degradation of selected proteins, including autophagy receptors. This pathway involved sorting of autophagy receptors into the intra-luminal vesicles of MVBs in a process that required the activity of the ESCRT-III component Charged multivesicular body protein 4b (CHMP4B) and the AAA ATPase VPS4, but seemed independent of VPS34 and the ULK1 complex [176]. Surprisingly, while ATG7 and ATG5 were dispensable for degradation of NBR1, Tax1-binding protein 1 (TAX1BP1), and Nuclear receptor coactivator 4 (NCOA4) through this pathway, endosomal microautophagy of p62 and NDP52 relied on ATG7/ATG5 as well as the LIR motif of p62, suggesting their binding to lipidated ATG8s [176]. It is tempting to propose that the rapid degradation of autophagy receptors by starvation-induced endosomal microautophagy prevents excessive and harmful selective macroautophagy under such acute conditions. Endosomal microautophagy was also reported in *Drosophila melanogaster* (fruit flies), but in contrast to the process described above it is initiated after prolonged starvation and targets substrates with a KFERQ motif in a Heat shock 70 kDa protein cognate 4 (Hsc70-4) dependent manner [177]. This process also requires MVB formation and ESCRT components, but proceeds independently of *Atg5, Atg7* and *Atg1* [177].

### 4.5. Functions of Non-Lipidated ATG8 Proteins

An increasing number of reports highlight functions of nonlipidated ATG8 proteins in processes not related to autophagy. It was recently shown that yeast Atg8 interacts with the vacuolar integral membrane protein Hfl1 (Has fused lysosomes 1) through a non-canonical AIM and that deletion of Hfl1 or Atg8 result in a tubular-shaped vacuole phenotype, which is not seen for Atg8 conjugation machinery mutants (atg3∆, atg4∆, atg5∆, and atg7∆), demonstrating a function of non-lipidated Atg8 [178]. Importantly, rescue of the vacuole phenotype specifically requires the interaction between Atg8 and Hfl1.

Another lipidation-independent role of LC3 was uncovered in ER-associated degradation (ERAD) [179]. ERAD is a cellular pathway responsible for the turnover of defective polypeptides in the ER, where unwanted products are retrotranslocated through the dislocon complex into the cytosol and targeted for proteasomal degradation. It was found that non-lipidated LC3 associates with ERAD tuning vesicles/EDEMosomes, which are ER derived vesicles that mediate clearance of ERAD regulators from the ER, including ER degradation-enhancing α-mannosidase-like 1 (EDEM1) and osteosarcoma amplified 9 (OS9) [179,180]. The role of LC3-I on EDEMosomes is still unclear, however it was found that Coronaviruses (CoVs) can hijack these vesicles for viral replication and that this requires LC3-I [180,181]. LC3 depletion inhibited replication of the CoVs Mouse Hepatitis Virus (MHV) and the equine arteritis virus (EAV), which could readily be reverted by re-introducing non-lipidatable LC3 to the cells. Moreover, Atg7 was dispensable for viral replication, further confirming a role of non-lipidated LC3 in EDEMosome-based viral replication.

## 5. Pathophysiological Roles of ATG8 Conjugation Machinery

Several model organisms have been used to study the pathophysiological roles of autophagy, using reverse genetic approaches to selectively delete specific core *ATG* genes. The discoveries made in such studies have greatly contributed to our understanding of this process in vivo. It is important to point out that some of the phenotypes seen in Atg8 conjugation-deficient models are different and sometimes less severe from those seen in model organisms lacking upstream *ATG* genes (e.g., *FIP200*, *Atg13*, *Atg9*, *Beclin1,* and *Vps34*), which could be due to other functions of the early *ATG* genes or the conjugation machinery not being absolutely essential for autophagy [10]. A large amount of in vivo studies have targeted components of the conjugation machinery in different model organisms (especially depletion or deletion of ATG5 or Atg7), which have been extensively reviewed elsewhere [2,10,182,183]. We will here mainly focus on the pioneering studies done in mice models lacking components of the ATG8 conjugation machinery and variants in corresponding human genes linked to disease.

### 5.1. Conjugation Machinery in Development

Mice lacking components of the Atg8 conjugation machinery (Atg7, Atg3, Atg12, Atg5, and Atg16L1) are born alive at the expected Mendelian frequency and appear almost normal at birth, but die within one day after birth [10]. This was initially interpreted as autophagy being dispensable for normal development, but it was later shown that autophagy plays a critical role in newly fertilized oocytes [184]. It was demonstrated that oocytes derived from oocyte-specific *ATG5* knockout mice failed to develop beyond the eight-cell stage when they were fertilized by *ATG5*-deficient sperm, but developed normally when fertilized by wild type sperm. Preimplantation autophagy seems to be important to support a normal rate of protein synthesis, but it could also be important for degradation of specific proteins or organelles at this stage. Interesting, elegant experiments in *Caenorhabditis elegans* early embryos demonstrated that sperm-derived mitochondria are degraded by selective autophagy, providing an explanation to the specific inheritance of maternal mtDNA [185].

The newborn *ATG* conjugation machinery-deficient mice that failed to survive the neonatal starvation period displayed suckling defects and had reduced concentrations of amino acids in plasma and tissues with signs of energy depletion [10]. Interestingly, *ATG5*-null mice rescued by *ATG5* expression in neurons were found to successfully suckle milk and survive for several months, demonstrating that neuronal autophagy is sufficient to maintain energy homeostasis and survival during early neonatal starvation [186]. However, these mice displayed inflammation in many tissues, atrophy in some organs and iron-deficiency anemia. Further corroborating the importance of neuronal autophagy for normal development, mice with neural-specific deletion of *ATG5* survived neonatal starvation, but displayed growth retardation and progressive motor and behavioral deficits, as well as accumulation of abnormal intracellular protein inclusions [187]. Similar results were obtained in mice with neuronal specific *ATG7* KO [188], showing that continuous autophagy-mediated clearance of cytosolic proteins prevents the accumulation of abnormal proteins, which can disrupt neural function and ultimately lead to neurodegeneration even in the absence of any disease-associated mutant proteins.

### 5.2. Conjugation Machinery in Disease

To investigate the functional significance of the ATG8 conjugation machinery and autophagy in adult mice, whole body ATG7 knockout mice was generated by tamoxifen-inducible conditional deletion of *ATG7* in adult mice [189]. These mice showed increased susceptibility to infection, extensive liver and muscle damage, and died of neurodegeneration two to three months after deleting *ATG7*. When subjected to 24 h fasting, these mice displayed extreme muscle wasting and died of hypoglycaemia, indicating that autophagy is required for glucose homeostasis and prevention of cachexia during fasting [189].

Several studies in autophagy-deficient *Drosophila* have confirmed the essential neuroprotective role of autophagy in adult brain. Flies lacking *Atg7* displayed neuronal death, shortened lifespan and neuronal aggregates [190]. Also, Atg5 mutant flies showed accumulation of protein aggregates in the brain and were characterized by severe movement disorders [191]. Interestingly, while flies lacking a functional Atg8a accumulated neuronal aggregates and had a shorter lifespan than control flies, driving overexpression of Atg8a in adult neurons was sufficient to significantly extend lifespan [192].

In line with an important role of autophagy in prevention of protein aggregation, several mutations in human ATG7 have been linked to neurodegenerative disease, including a V471A polymorphism in ATG7 that correlates with an earlier disease onset in Huntington disease populations [193,194]. Moreover, genetic variants were identified in the ATG7 gene promoter of patients with Parkinson’s disease, suggesting that altered transcriptional activity of ATG7 may be a risk factor [195]. Furthermore, a causative missense mutation in *ATG5* was identified in two siblings with congenital ataxia, mental retardation, and developmental delay [191].

The role of autophagy in cancer is complicated and diverse. Conditional autophagy-deficient mouse models develop spontaneous liver tumors [196], indicating that autophagy is important for the suppression of spontaneous tumorigenesis. However, autophagy may also facilitate growth of already formed tumors and cause cancer cells to become more resistant to chemotherapy, suggesting that inhibition of autophagy could be beneficial against cancer. The latter hypothesis was tested in the above-mentioned whole body tamoxifen-induced ATG7 KO mice model. Importantly, deletion of *ATG7* in mice with pre-existing non-small cell lung cancer (NSCLC) stopped further tumor growth and promoted tumor cell death, before the destruction of normal tissues were detected [189], indicating that transient inhibition of autophagy may be therapeutically beneficial in cancer.

An autophagy- and ATG7-independent role of ATG5 in cancer was recently demonstrated. ATG5 was found to enhance the migration and metastasis of breast cancer cells by regulation of exosome release [197]. Exosomes are extracellular vesicles that are derived from the intraluminal vesicles of MVBs when they fuse with the plasma membrane. In contrast to the ATG12–ATG3 conjugate, required for budding of intraluminal vesicles into MVBs [73], ATG5 did not affect MVB formation. Rather, ATG5 was found to regulate sorting of the V-ATPase subunit ATP6V1E1 into the intraluminal vesicles along with LC3, thereby decreasing the acidification of MVB, which promoted exosome release [197]. While LC3-II was detected within exosomes in wild type cells, LC3-I was found in exosomes in ATG7 deficient cells and LC3 was absent from exosomes in ATG5 and ATG16L1 cells, indicating that LC3 lipidation is not a prerequisite for LC3 sorting into exosomes.

### 5.3. Role of Conjugation Machinery in Immunity

Several studies have demonstrated an important role of autophagy in immunity and inflammation (for a review see [198,199]). Autophagy is able to fight invading pathogens, but also balances the activation of the immune system to avoid excessive inflammation. Several genome wide association studies (GWASs) have identified single nucleotide polymorphisms (SNPs) in *ATG5* that seem to predispose for the autoimmune disease systemic lupus erythematosus (SLE) and lead to increased ATG5 levels in SLE patients [200,201,202,203,204].

A SNP in ATG16L1 (T300A) is a major risk variant for Crohn’s disease (CD), an inflammatory bowel disease (IBD). Mice harboring the ATG16L1 T300A mutation exhibit decreased antibacterial autophagy and abnormal Paneth cell lysozyme granule distribution [205,206]. The T300A mutation is just adjacent to a caspase 3 cleavage site (aa 296–299) and it was found that the mutation increased susceptibility for ATG16L1 cleavage [207]. Interestingly, ULK1-mediated phosphorylation of ATG16L1 S278 appears to promote specific cleavage of the ATG16L1 T300A mutant protein, while enhancing the function of wild type ATG16L1 in xenophagy [79]. As the ULK1 phosphosite (S278) is missing in the ATG16L1 α isoform (missing aa 266–284 of the β isoform) it is exciting to note that the wild type version of both ATG16L1 isoforms seemed to be cleaved at similar rates [207], indicating that ULK1-mediated S278 phosphorylation specifically targets degradation of the CD-associated ATG16L1 T300A variant. Recently it was reported that cells harboring the ATG16L1 T300A allele were unable to perform plasma membrane repair in response to membrane damage induced by the *Listeria monocytogenes* toxin listeriolysin O, providing a possible explanation for the association of this variant to IBD [208].

As several recent studies have demonstrated autophagy-independent roles of ATG8 proteins and their conjugation machinery, it will be important to clarify if the developmental and pathological defects observed in conjugation machinery-deficient mice are caused by dysfunctional autophagy or linked to autophagy-independent functions of these proteins. The WD domain-containing C-terminus of ATG16L1 is dispensable for canonical autophagy [209] and in line with this, it was found that mice lacking this part of Atg16L1 were capable of activating autophagy, but unable to activate LAP [210]. These mice grew at the same rate as littermate controls, were fertile and did not have obvious defects in liver, kidney, brain or muscle homeostasis, indicating that autophagy maintains tissue homeostasis in mice independently of LAP [210]. In contrast, when the WIPI2 binding site (E226 and E230) was also deleted (in addition to the C-terminal WD repeats) the mice appeared defective in both autophagy and LAP in all tissues, and although they survived post-natal starvation, they were smaller and died around six months of age. Thus, systemic deletion of conjugation machinery genes is more severe than deletion of the ATG16L1 [210]. The LAP-deficient mice displayed a significant incapacity of bone marrow derived dendritic cells (BMDCs) to present exogenous antigens compared to wild-type BMDCs [41]. The mechanisms underlying the role of the ATG16L1 WD domain in LAP remains elusive, but might be linked to the reported binding of the WD domain of ATG16L1 to Nucleotide-binding oligomerization domain-containing protein (NOD)-like receptors [211], Mediterranean fever (MEFV)/Tripartite motif (TRIM)-containing protein 20 (TRIM20) [212], Transmembrane protein 59 (TMEM59) [213] and Protein eva-1 homolog A (EVA1A)/TMEM166 [214].

### 5.4. Pathogen-Mediated Modifications of the ATG8 Conjugation Machinery

Our immune system is constantly challenged by invading pathogens that try to find ways to escape or inhibit being targeted for degradation. As described above, selective pathogen degradation (xenophagy), as well as various pathways involving non-conventional roles of ATG8 proteins, are important for our first line of defense against infection (innate immunity). Thus, many pathogens have found ways to escape these pathways by inhibiting ATG8 proteins and their conjugation machinery. In contrast, some pathogens take advantage of the autophagy machinery for their own benefit.

The invasive bacteria *Shigella* avoids detection by autophagy through secretion of the type III secretion system (T3SS) effector protein IcsB, which surrounds the bacteria and prevents interaction of the *Shigella* protein VirG with ATG5 and subsequent recruitment of autophagy receptors and LC3 [215,216]. Similarly, the *Salmonella* T3SS effector SopF blocks autophagy by ADP-ribosylation of Q124 of the V-ATPase subunit ATP6V0C, blocking recruitment of ATG16L1 to the bacteria-containing vacuole and subsequent degradation of bacteria by autophagy [159]. *Listeria monocytogenes* expresses the actin polymerization inducing protein ActA, preventing its recognition by the autophagy machinery [217,218,219]. Interestingly, it also induces mitophagy, which seems to facilitate bacterial survival by reduction of the level of ROS [220]. The intracellular pathogen *Legionella pneumophila* interferes with autophagy by expressing the bacterial effector protein RavZ, which directly deconjugates lipidated LC3 and GABARAP proteins [221]. RavZ cleaves N-terminal of the carboxyl-terminal glycine residue, generating non-lipidatable LC3/GABARAP proteins and a novel PE substrate with a glycine modification. The truncated LC3/GABARAP proteins cannot be further processed by ATG7 and ATG3, thereby irreversibly inactivating them during infection [221].

While autophagy plays an important anti-viral function, several viruses successfully avoid autophagy and even hijack the autophagy machinery for their own replication. Influenza A virus hijacks autophagy via interaction of a LIR in the cytoplasmic tail of its M2 proton channel with LC3, promoting LC3 localization to the plasma membrane [222]. Mutations in the M2 LIR motif interfere with filamentous budding and reduce virion stability. Moreover, LC3 relocalization depends on the proton channel activity of M2 and the WD domain of ATG16L1, suggesting that the LC3 lipidation could be triggered by loss of cellular pH gradients. Thus, hijacking autophagy probably facilitates transmission of *Influenza* A virus infection between organisms by enhancing the stability of viral progeny.

## 6. Concluding Remarks

The ATG8 proteins have been widely used as markers to study autophagy for two decades and have been imperative for our understanding of mechanisms involved in regulation and execution of the pathway. However, their exact function in autophagy are only starting to become clear. We know that although autophagosomes can form in the absence of lipidated ATG8 proteins, GABARAP proteins seems to be particularly important for membrane recruitment of core autophagy machinery and for autophagosome closure, while LC3 proteins may be specifically involved in cargo recruitment. Several structure–function studies have provided important information about how such specificity is obtained by binding of LC3 and GABARAP proteins to proteins containing specific LIR motifs. However, the molecular mechanisms involved in regulation of the localization and function of the different LC3 and GABARAP proteins in time and space in vivo still remains largely unknown. It is also not clear to what degree the LC3 and GABARAP subfamilies have specific and/or redundant roles in autophagy. Another open question is whether the role of LC3/GABARAP proteins in autophagosome biogenesis differ depending on the autophagy-inducing signal and the presence of cargo or not. Finally, we are only beginning to understand the non-conventional roles of LC3 and GABARAP proteins in pathways not related to autophagy. Clearly, the latter will be an important field of research within the near future, but it is also important that autophagy researchers keep this in mind when targeting ATG8 proteins or their conjugation machinery.

## Figures and Tables

**Figure 1 cells-08-00973-f001:**
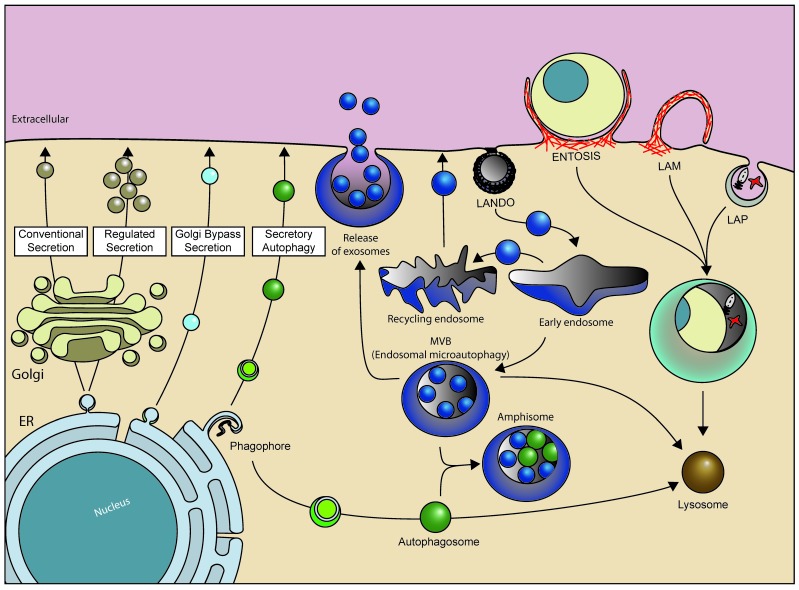
Overview of conventional and several non-conventional pathways where autophagy-related (ATG)8 proteins and their conjugation machinery are utilized. Briefly, ATG8 protein conjugation is important during macroautophagy where it facilitates cargo recruitment, closure of the autophagosome and degradation of the inner membrane. ATG8 proteins and their machinery are also shown to affect protein secretion, including regulated secretion (Lysozyme, Glut4), Golgi-bypass secretion (CFTR), and secretory autophagy, which includes secretion of cytosolic interleukin 1 beta (IL-1β) and insulin-degrading enzyme (IDE). ATG8 proteins are found to be important for maturation of compartments following light chain 3 (LC3)-associated phagocytosis (LAP), entosis, micropinocytosis, and LC3-associated endocytosis (LANDO). Core components of the conjugation machinery have further been implicated in endosomal microautophagy and exosome release.

**Figure 2 cells-08-00973-f002:**
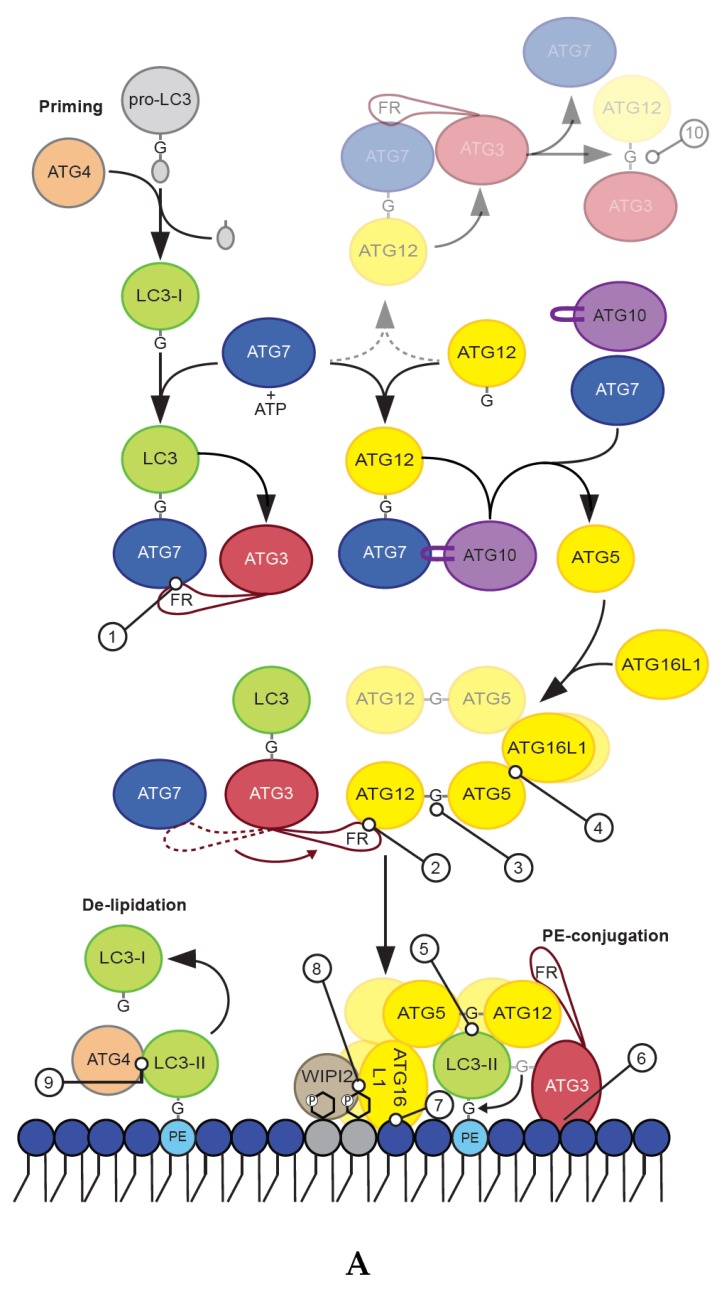
(**A**) Mechanisms and interactions (protein and membrane) of core autophagy components required for priming, lipidation and de-lipidation of ATG8 proteins during starvation-induced autophagy are illustrated. The illustration also includes the conjugation of ATG12 to ATG5 as well as the less commonly reported conjugation of ATG12 to ATG3 (shaded), which serves its purpose outside of autophagy. Key interactions are numbered and shown in greater detail in (**B**). (**B**) ATG3 interacts with ATG7 and ATG12 through motifs located in a flexible region (aa 88–192), these motifs are referred to as RIA7 (1) and RIA12 (2) respectively. An amphipathic helix in the very N-terminal end of ATG3 (aa 1–26) provides membrane binding essential for lipidation of ATG8 proteins (6). G140 of ATG12 is conjugated to K130 of ATG5 (3) or K243 of ATG3 (10). It is proposed that V62 and W139 of ATG12 (5) serves as a LIR motif to secure the complex to the membrane of an autophagosome during lipidation. ATG16L1 interacts with ATG5 through its N-terminal helix (aa 13–28) (4). Membrane interaction of ATG16L1 is achieved through multiple interactions; an amphipathic helix near the N-terminal end provide membrane binding and is required for ATG8 lipidation (7), conserved residues in the coiled-coil domain together with its interaction with the phosphatidylinositol 3-phosphate (PtdIns(3)P) effector protein WIPI2 (8) provide binding to PtdIns(3)P and membrane binding through the β-isoform specific insert is important for ATG8 protein lipidation to damaged endosomes/lysosomes. The β-isoform specific insert is a highly post-translationally modified area in ATG16L1 and includes a confirmed phosphorylation site for ULK1 on S278. ATG4B interacts with ATG8 proteins through its catalytic core (CC) and an N-terminal and a C-terminal LC3-interacting region (LIR) (9).

**Table 1 cells-08-00973-t001:** Known interactions and functions of the core ATG8 conjugation machinery proteins during priming, lipidation/conjugation and de-lipidation.

Protein	Main Function	Interactions or Site-Specific Actions	Residues Important for Interaction or Function	Refrences
ATG7	E1 like enzyme	ATG3	R246D, W243A	[23]
		mATG8	Catalytic cysteine: C572	[24]
ATG3	E2 like enzyme	ATG7	RIA7: aa 157–181	[23]
		ATG12	RIA12: aa 140–170	[25]
			Direct conjugation: K243	[26]
		mATG8	Catalytic cystine: C264	[27]
		Membrane	Amphipathic helix: aa 1–26	[28]
		Caspase cleavage	L166, E167, T168, D169 ^V^ E170	[29]
ATG10	E2 like enzyme	ATG12	Catalytic cysteine: C166	[30]
ATG12	Component of E3 like complex	ATG3	Interaction with flexible region: K54, K72, and W73 and more	[31]
			Direct conjugation: G140	[26]
		ATG5	Direct conjugation: G140	[32]
		mATG8	Potential LIR: V62, W139	[33]
ATG5	Component of E3 like complex	ATG12	Direct conjugation: K130	[32]
		ATG16L1	T249, P250, W253, V7, I243, P245, T249, P250, W253, L258, H241, D10	[34]
ATG16L1	Component of E3 like complex	ATG5	AFIM: W13, I17, L21, R24, Q28	[34]
		WIPI2	E226 and E230	[35]
		RB1CC1/FIP200	aa 235–241	[35,36,37,38]
		Membrane	Amphipathic helix: aa 28–44	[39]
			PtdIns(3)P interaction: I171, K179 and R193	[40]
			β-isoform insert: aa 266–284	[39]
		Phagosome recruitment (LAP)	Required for LAP: F467, K490(Interaction partner not identified)	[41]
ATG4A	mATG8 cysteine proteases	mATG8	LIR: F393, E394, I395, L396	[42]
		mATG8-cleavage	Catalytic triad: C77/D279/H281	[43,44]
			mATG8-I processing: GABARAP, GABARAPL1, GABARAPL2 (LC3A and LC3C not tested)	[15,19,20,45]
			mATG8-II processing: GABARAPL1 and GABARAPL2 (LC3A not tested)	[15,19,46]
ATG4B	mATG8 cysteine proteases	mATG8	N-terminal LIR: Y8, D9, T10, L11	[47]
			C-terminal LIR: F388, E389, I390, L391	[42]
		mATG8-cleavage	Catalytic triad: C74, D278 and H280	[48]
			mATG8-I processing: LC3B, LC3C, GABARAP, GABARAPL1, GABARAPL2 (LC3A not tested)	[15,19,45,46]
			mATG8-II processing: LC3B, GABARAP, GABARAPL1, GABARAPL2 (LC3A not tested)	[15,19,46]
ATG4C	mATG8 cysteine proteases	mATG8-cleavage	Catalytic cystine (prediction): C111/D345/H347	[49]
			mATG8-I processing: No processing shown (LC3A, LC3C and GABARAP not tested)	[19,45]
			mATG8-II processing: GABARAPL2 (LC3A, LC3C and GABARAP not tested)	[19]
		Caspase cleavage	D7, E8, V9, D10 ^V^ K11	[50]
		mATG8-processing post caspase cleavage (∆ aa 1–10)	mATG8-I processing: No processing shown (LC3A, LC3C and GABARAP not tested)	[19]
			mATG8-II processing: LC3B, GABARAPL1, GABARAPL2 (LC3A, LC3C and GABARAP not tested)	[19]
		Predicted MTS	aa 11–40	[51]
ATG4D	mATG8 cysteine proteases	mATG8-cleavage	Catalytic cystine (prediction): C134/A356/H358	[49]
			mATG8-I processing: (LC3A, LC3C and GABARAP not tested)	[19,45]
			mATG8-II processing: (LC3A, LC3C and GABARAP not tested)	[19]
		Caspase cleavage	D60, E61, V62, D63 ^V^ K64	[50]
		mATG8-processing post caspase cleavage (∆ aa 1–63)	mATG8-I processing: (LC3A, LC3C and GABARAP not tested)	[19]
			mATG8-II processing: LC3B, GABARAPL2 (LC3A, LC3C and GABARAP not tested)	[19]
		Predicted MTS	aa 64–105	[51]

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
