# Peer review of "Mechanisms and Pathophysiological Roles of the ATG8 Conjugation Machinery"

_cells, 2019, doi:10.3390/cells8090973_

Round 1
Reviewer 1 Report
This manuscript summarizes current knowledge on the mechanisms and pathophysiological roles of the ubiquitin-like ATG8 conjugation machinery. The manuscript is well-written and covers recent critical studies extensively and thus will be beneficial for the readers of Cells. Some key findings that were initially reported in yeast studies are lacking, which should be cited in order to raise the level of completion of the manuscript (listed in minor points below).
Minor points
1) line 58, “phosphatidylethanolamine (PE) the autophagic membrane”, insert “in” after PE.
2) lines 59-60, cite references for the usage of ATG8 as autophagosomal markers.
3) lines 152-153, ATG13 FR was first reported for yeast Atg13 (PMID 17227760) and thus should also cite this paper.
4) line 269, add citation (PMID 23392225) to the ATG16L1-FIP200 interaction.
5) line 400, add citation (PMID 18508918) that describes the relationship between Atg8 expression level and autophagosomal size.
6) line 460, the first identified autophagy receptor is Atg19 and thus describe it and cite the following two papers (PMID 11430817, 11382752).
7) lines 470-471, the LIR consensus motif, W-x-x-L, was first discovered by structural comparison of Atg8-Atg19 and LC3-p62 complexes (PMID 19021777) and thus cite this paper.
8) 4. Non-conventional roles of ATG8s …”, in yeast, Atg8 was reported to function in non-autophagy processes without conjugation with PE (for example, PMID 30451685). It would be better to include such non-conventional role of ATG8 without conjugation in this chapter.
9) line 540, “single layered membrane” is confusing because it could mean single-layered phospholipid observed in lipid droplets.
10) line 770, RavZ cleaves the peptide bond at the N-terminus of the carboxy-terminal glycine residue. From the current sentence, we do not know which side of glycine is cleaved.
11) Figure 1 legend, “non-conventional pathways were ATG8”, “were” should be “where”.
Author Response
Reviewer 1
This manuscript summarizes current knowledge on the mechanisms and pathophysiological roles of the ubiquitin-like ATG8 conjugation machinery. The manuscript is well-written and covers recent critical studies extensively and thus will be beneficial for the readers of Cells. Some key findings that were initially reported in yeast studies are lacking, which should be cited in order to raise the level of completion of the manuscript (listed in minor points below).
We thank the reviewer for the positive review and helpful comments, which have all been addressed in the text as described below.
Minor points
1) line 58, “phosphatidylethanolamine (PE) the autophagic membrane”, insert “in” after PE.
This has been corrected.
2) lines 59-60, cite references for the usage of ATG8 as autophagosomal markers.
The proper references for Atg8/LC3 as autophagosomal markers are included at the end of the prior sentence “…ATG8 proteins become directly conjugated to the lipid phosphatidylethanolamine (PE) in the autophagic membranes and remain bound throughout the pathway” In addition, we have now inserted a reference to the Autophagy guidelines paper (PMID:26799652) for the usage of ATG8s as autophagosomal markers.
3) lines 152-153, ATG13 FR was first reported for yeast Atg13 (PMID 17227760) and thus should also cite this paper.
The PMID 17227760 (Yamada et al, J Biol Chem 2007) citation has now been included.
4) line 269, add citation (PMID 23392225) to the ATG16L1-FIP200 interaction.
The PMID 23392225 (Nishimura et al, EMBO Rep 2013) citation has now been included.
5) line 400, add citation (PMID 18508918) that describes the relationship between Atg8 expression level and autophagosomal size.
The PMID 18508918 (Xie et al, Mol Biol Cell 2008) citation has now been included.
6) line 460, the first identified autophagy receptor is Atg19 and thus describe it and cite the following two papers (PMID 11430817, 11382752).
We have included the following sentence: “Yeast Atg19 was the first autophagy receptor identified, being important for selective targeting of a precursor form of protein aminopeptidase I (prAPI) to the vacuole in a process referred to as the cytoplasm-to-vacuole (Cvt) pathway [115-117].” In addition to the two suggested references we have cited Shintani, T., et al., Mechanism of cargo selection in the cytoplasm to vacuole targeting pathway.Dev Cell, 2002. 3(6): p. 825-37.
7) lines 470-471, the LIR consensus motif, W-x-x-L, was first discovered by structural comparison of Atg8-Atg19 and LC3-p62 complexes (PMID 19021777) and thus cite this paper.
We decided to cite a review article by Terje Johansen as a reference for the consensus LIR sequence, as this is based on the work of several groups, where Johansen had the first paper describing a LIR (Pankiv et al, 2007), but we agree that we should have cited PMID 19021777 for the reference to the hydrophobic pockets on the LC3/GABARAP proteins. This citation has now been included.
8) 4. Non-conventional roles of ATG8s …”, in yeast, Atg8 was reported to function in non-autophagy processes without conjugation with PE (for example, PMID 30451685). It would be better to include such non-conventional role of ATG8 without conjugation in this chapter.
We agree with the reviewer that examples of non-lipidated Atg8 functions should be included in this chapter. We have now included the following text:
4.5. Functions of non-lipidated ATG8 proteins
An increasing number of reports highlight functions of nonlipidated ATG8 proteins in processes not related to autophagy. It was recently shown that yeast Atg8 interacts with the vacuolar integral membrane protein Hfl1 (Has fused lysosomes 1) through a non-canonical AIM and that deletion of hfl1or atg8result in a tubular-shaped vacuole phenotype, which is not seen for Atg8 conjugation machinery mutants (atg3∆, atg4∆, atg5∆ and atg7∆), demonstrating a function of non-lipidated Atg8 [173]. Importantly, rescue of the vacuole phenotype specifically requires the interaction between Atg8 and Hfl1.
Another lipidation-independent role of LC3 was uncovered in ER‐associated degradation (ERAD) [174]. ERAD is a cellular pathway responsible for the turnover of defective polypeptides in the ER, where unwanted products are retrotranslocated through the dislocon complex into the cytosol and targeted for proteasomal degradation. It was found that non-lipidated LC3 associates with ERAD tuning vesicles/EDEMosomes, which are ER derived vesicles that mediate clearance of ERAD regulators from the ER, including ER degradation‐enhancing α‐mannosidase‐like 1 (EDEM1) and osteosarcoma amplified 9 (OS9) [174, 175]. The role of LC3-I on EDEMosomes is still unclear, however it was found that Coronaviruses (CoVs) can hijack these vesicles for viral replication and that this requires LC3-I [175, 176]. LC3 depletion inhibited replication of the CoVs Mouse Hepatitis Virus (MHV) and the equine arteritis virus (EAV), which could readily be reverted by re-introducing non-lipidatable LC3 to the cells. Moreover, Atg7 was dispensable for viral replication, further confirming a role of non-lipidated LC3 in EDEMosome-based viral replication.
9) line 540, “single layered membrane” is confusing because it could mean single-layered phospholipid observed in lipid droplets.
We have now changed this to “Lipidation of ATG8s to single-membrane compartments”
10) line 770, RavZ cleaves the peptide bond at the N-terminus of the carboxy-terminal glycine residue. From the current sentence, we do not know which side of glycine is cleaved.
We have changed the text to read: “RavZ cleaves N-terminal of the carboxyl-terminal glycine residue, generating non-lipidatable LC3/GABARAP proteins and a novel PE substrate with a glycine modification.»
11) Figure 1 legend, “non-conventional pathways were ATG8”, “were” should be “where”.
This has now been corrected.
Reviewer 2 Report
The review by Lystad et al is a comprehensive description of the mammalian Atg8 homologue protein families and their conjugation systems. The authors further describe the state-of-the-art knowledge of the autophagic and non-autophagic functions of these proteins. Further chapters nicely summarize the relevance of these components in diseases, xenophagy and immunology.
The review is balanced, easy to read and covers the literature. I strongly recommend its publication.
Minor comments:
Citations in line 55 are not in journal style and do not appear in the references. In Fig.2 and the text the binding of Atg16L1 to PI3P is highlighted. It should become clearer that WIPI2 is also a strong PI3P binding adaptor. Line 182 missing blank Line 379 blue text Line 506, there is a double blank line 604 missing blank
Author Response
Reviewer 2
The review by Lystad et al is a comprehensive description of the mammalian Atg8 homologue protein families and their conjugation systems. The authors further describe the state-of-the-art knowledge of the autophagic and non-autophagic functions of these proteins. Further chapters nicely summarize the relevance of these components in diseases, xenophagy and immunology.
The review is balanced, easy to read and covers the literature. I strongly recommend its publication.
We thank the reviewer for these encouraging remarks to our manuscript. All comments and concerns have been addressed in the text as described below.
Minor comments:
Citations in line 55 are not in journal style and do not appear in the references.
This has now been corrected.
In Fig.2 and the text the binding of Atg16L1 to PI3P is highlighted. It should become clearer that WIPI2 is also a strong PI3P binding adaptor.
We have now added the following text to the Fig.2 legend: “…together with its interaction with the PtdIns(3) effector protein WIPI2 (8)….”. We have also included the following text to chapter 2.2.3: “ATG16L1 seem to specify the site of ATG8 lipidation [44], which is likely mediated by its interaction with specific membrane-bound proteins, including the PtdIns(3)P effector WIPI2b [47], the ULK1 complex component FIP200 [48] and ubiquitin [49].”
Line 182 missing blank
This has been corrected.
Line 379 blue text
Corrected.
Line 506, there is a double blank
Corrected.
Line 604 missing blank
Corrected.